# Curated multiple sequence alignment for the Adenomatous Polyposis Coli (*APC*) gene and accuracy of *in silico* pathogenicity predictions

**Alexander D. Karabachev**[1], **Dylan J. Martini**[1¤a], **David J. Hermel**[1¤b], **Dana Solcz**[1], **Marcy E. Richardson**[2], **Tina Pesaran**[2], **Indra Neil Sarkar**[3,4], **Marc S. Greenblatt**[1] *

1 Department of Medicine, University of Vermont, Larner College of Medicine, Burlington, Vermont, United States of America, 2 Ambry Genetics, Aliso Viejo, California, United States of America, 3 Center for Biomedical Informatics, Brown University, Providence, Rhode Island, United States of America, 4 Rhode Island Quality Institute, Providence, Rhode Island, United States of America

¤a Current address: Emory University School of Medicine, Atlanta, Georgia, United States of America
¤b Current address: Scripps Clinic, La Jolla, California, United States of America
* Marc.Greenblatt@uvmhealth.org

**Data Availability Statement:** The data underlying the results presented in the study are either within the manuscript or available through ClinVar at the

## Abstract

Computational algorithms are often used to assess pathogenicity of Variants of Uncertain Significance (VUS) that are found in disease-associated genes. Most computational methods include analysis of protein multiple sequence alignments (PMSA), assessing interspecies variation. Careful validation of PMSA-based methods has been done for relatively few genes, partially because creation of curated PMSAs is labor-intensive. We assessed how PMSA-based computational tools predict the effects of the missense changes in the *APC* gene, in which pathogenic variants cause Familial Adenomatous Polyposis. Most Pathogenic or Likely Pathogenic APC variants are protein-truncating changes. However, public databases now contain thousands of variants reported as missense. We created a curated APC PMSA that contained >3 substitutions/site, which is large enough for statistically robust *in silico* analysis. The creation of the PMSA was not easily automated, requiring significant querying and computational analysis of protein and genome sequences. Of 1924 missense APC variants in the NCBI ClinVar database, 1800 (93.5%) are reported as VUS. All but two missense variants listed as P/LP occur at canonical splice or Exonic Splice Enhancer sites. Pathogenicity predictions by five computational tools (Align-GVGD, SIFT, PolyPhen2, MAPP, REVEL) differed widely in their predictions of Pathogenic/Likely Pathogenic (range 17.5–75.0%) and Benign/Likely Benign (range 25.0–82.5%) for *APC* missense variants in ClinVar. When applied to 21 missense variants reported in ClinVar and securely classified as Benign, the five methods ranged in accuracy from 76.2–100%. Computational PMSA-based methods can be an excellent classifier for variants of some hereditary cancer genes. However, there may be characteristics of the *APC* gene and protein that confound the results of *in silico* algorithms. A systematic study of these features could greatly improve the automation of alignment-based techniques and the use of predictive algorithms in hereditary cancer genes.

NCBI https://www.ncbi.nlm.nih.gov/clinvar/?term=APC%5Bgene%5D.

**Funding:** INS U54 GM115677 National Institutes of Health The funders had no role in study design, data collection and analysis, decision to publish, or preparation of the manuscript.

**Competing interests:** I have read the journal's policy and the authors of this manuscript have the following competing interests: MER and TP are employees of Ambry Genetics, Inc. No other authors have competing interests.

# Introduction

Multi-gene panel testing is now routine for identifying hereditary cancer susceptibility, leading to increased detection of pathogenic mutations, which can improve clinical management. However, testing often identifies variants of uncertain significance (VUS), which are often missense amino acid (AA) substitutions, small in frame deletions and duplications, or non-coding changes [1, 2]. VUS in genes that predispose to hereditary cancer and other disorders are rapidly accumulating in variant databases. For example, the ClinVar database at the National Center for Biotechnology Information at the United States National Library of Medicine provides a freely accessible archive of variants with assertions regarding the pathogenicity of each variant with the indicated phenotype from submitting laboratories and expert panels [3]. The classification of these VUS represents a major challenge in clinical genetics.

Computational *(in silico)* tools have been developed to help predict whether or not the protein function will be disrupted (reviewed in [4]). *In silico* tools often use Protein Multiple Sequence Alignments (PMSA) to consider the evolutionary conservation and biophysical properties of the wild type and variant protein to make predictions of pathogenicity. PMSA-based computational methods are complicated to use properly (reviewed in [4]). The PMSA must be of high quality and sample enough species to provide reliable data [5, 6]. These *in silico* methods have been validated for relatively few hereditary cancer genes in which pathogenic missense variants are not rare (BRCA1/2, the mismatch repair [MMR] genes, TP53, a few others) [5, 7–11]. They have not often been validated for other genes, and for some genes predictive value was not strong [12]. However, they are often cited as evidence in favor or against pathogenicity of variants for genes in which validation is lacking. The American College of Genetics and Genomics (ACMG) and the Association for Molecular Pathology (AMP) published guidelines for evaluating the pathogenicity of variants in Mendelian disease genes, including general rules for the use of *in silico* tools [13].

Missense pathogenic variants are rare in some genes, including *APC*, the gene responsible for Familial Adenomatous Polyposis (FAP). *APC* has been sequenced frequently in clinical genetic testing, but few missense pathogenic variants have been identified, for reasons that have not been clearly demonstrated [14]. The increase in clinical DNA sequencing tests for cancer predisposition has led to an increase in missense VUSs in APC that require classification.

Here we systematically apply *in silico* methods to *APC*, assessing the logistics and results of using these commonly available tools to predict pathogenicity of missense variants in a gene for which missense is an uncommon mechanism of pathogenicity.

# Materials and methods

Sequence and variant data are publically available from databases at the NLM. The study protocol was determined to be exempt from human subject regulations by Western IRB, as the data were de-identified.

## APC sequences and multiple sequence alignments, phylogenetic analysis

Amino acid sequences were collected by searching NCBI's online Gene database (http://www.ncbi.nlm.nih.gov/gene), for "APC" in 2013, 2015, and 2018. PMSAs were made using Clustal Omega from the European Bioinformatics Institute (EBI) (https://www.ebi.ac.uk/Tools/msa/clustalo) and MUSCLE v3.8.31 [15] and examined using Mesquite, a software for evolutionary biology (http://mesquiteproject.wikispaces.com/) [16].

Misaligned areas were manually adjusted after the MUSCLE alignment. Gaps and insertions in the PMSA were analyzed to determine if the sequences in question were likely true

indels or likely to be artifacts of computer analysis of genome annotation. BLAST searches were performed of inserted runs of AAs that did not align with any other species in our PMSA, using Protein BLAST, with default settings and query sequences of minimum length 30. For a "positive BLAST", the sequence results needed to show the presence of either homologs of the query sequence in APC from other organisms or from known protein domains. For a "negative BLAST", the only result was the sequence from the species used in the search query. Exon boundaries were identified using the NCBI Gene Database. If an entire exon from one species did not align with the other sequences and was deemed BLAST negative, that exon was removed from the PMSA, using the rationale that it would be irrelevant to a variant found in humans.

Phylogenetic trees were constructed from the curated APC alignment using a Maximum Parsimony-based method implemented in PAUP* (Phylogenetic Analysis Using Parsimony [*and Other Methods]), Version 4, Maximum Likelihood [17, 18], and Bayesian method as implemented in MrBayes [19].

Nucleotide regions flanking prospective indels were analyzed using two splice site calculators: (1) SpliceSiteFrame, (http://ibis.tau.ac.il/ssat/SpliceSiteFrame.htm), a splice site calculator from Tel Aviv University, and (2) the online tool from the GENIE program [20] (http://rulai.cshl.edu/new_alt_exon_db2/HTML/score.html), The maximum 3' score for a perfect splice site would be 14.2, and the score for a perfect 5' splice score would be 12.6; these rarely occur. Average scores for the 3' and 5' sites are 7.9 and 8.1 respectively.

**Substitution per site.** Absolute conservation of an amino acid in a PMSA can be determined with statistical significance ($P<0.05$) if the PMSA contains at least three substitutions per site (subs/site, i.e., three times as many variants among all sequences as there are codons in the gene [5, 6]. In order to determine if APC alignments contained three subs/site, we used the PHYLIP (Phylogeny Inference Package) version 3.6a2 ProtPars program form the University of Washington, Department of Genetics (http://evolution.genetics.washington.edu/phylip.html), with the alignment converted to PHYLIP format. To convert the alignment from Clustal Omega format to PHYLIP format and all other formats used during the analyses, the EMBOSS Seqret from EBI (https://www.ebi.ac.uk/Tools/sfc/emboss_seqret/) and Mesquite Version 3.51 tools were used (https://www.mesquiteproject.org/).

## Predictions of effects of APC missense substitutions

In July 2013, 46 APC missense variants were collected from the LOVD database maintained by the International Society for Gastrointestinal Hereditary Tumors (InSiGHT). On May 30, 2018, 4891 variants observed by clinical genetic testing were collected from the ClinVar database (http://www.ncbi.nlm.nih.gov/clinvar/).

**Computational algorithms.** The pathogenicity of each missense variant recorded in ClinVar was predicted using the programs Align-GVGD, SIFT, PolyPhen2 MAPP, and REVEL. *AlignGVGD* uses PMSAs and the biophysical properties of amino acid substitutions to calculate the range of variation at each position. Each variant is assigned a grade of C65 to C0 representing decreasing probability of deleterious, with C0 representing likely neutral AA substitutions [21]. (http://agvgd.hci.utah.edu/about.php).

*SIFT (Sorting Intolerant From Tolerant)* creates position specific scoring matrices derived from PMSAs. Each missense substitution predicted as "Tolerated' or "Affects Protein Function" [22]. (http://sift.bii.a-star.edu.sg/).

*PolyPhen2* combines its own pre-built sequence alignment with protein structural characteristics, calculating a score used to classify each variant into three categories: benign, possibly

damaging and probably damaging. (http://genetics.bwh.harvard.edu/pph2/index.shtml) [23]. We combined the categories of "possibly damaging" and "probably damaging".

***MAPP (Multivariate Analysis of Protein Polymorphisms)*** also combines a PMSA with the physiochemical characteristics of each AA position, predicting which AA should be deleterious and which should be neutral at each position in the PMSA [24] (http://www.ngrl.org.uk/Manchester/page/mapp-multivariate-analysis-protein-polymorphism).

***REVEL (Rare Exome Variant Ensemble Learner)*** [25] is an ensemble method that uses machine learning to combine the results of 13 individual predictors, using independent test sets that did not overlap with sets used to train its component features. REVEL output classes were designated as "Deleterious" for variants with a REVEL score $\geq 0.5$ and "Neutral" with a REVEL score $< 0.5$ [25].

## Results

### PMSA creation

Results from searching the NCBI Gene database for "APC" initially yielded reliable full length APC protein sequences from 38 organisms. We encountered a number of challenges to the simple automated assembly of a meaningful APC PMSA, including:

a.  *Large inconsistencies with the APC human sequence.* In order to include only sequences which accurately reflect human biology, such sequences were omitted.

b.  *Multiple APC isoforms were found for 21 organisms.* To choose the most appropriate isoform, all 104 sequences were aligned using Clustal2W. Isoforms that lacked a common beginning protein sequence of MAA were deleted (N = 26). When duplicate sequences were found for the same species, the more complete sequence was used, and if similar length isoforms of the same organism were found with a common sequence initiation, the lowest number isoform was chosen.

c.  *Large deletions or insertions.* Many of these could easily be identified as errors in automated identification of exon-intron boundaries. In most cases we could identify the appropriate boundary and either insert or delete the appropriate sequence. For insertions that were unique to one organism, especially in areas of otherwise high homology, BLAST was used to seek other homologues of the inserted sequence, and assessed the relevant nucleotide sequence for plausible overlooked splice sites.

d.  *Small deletions or insertions.* Short gaps that were confirmed to occur distant from an exon-intron boundary were allowed. The longest such gap was AA 1631–1637 in *Loxodonta africana* (African elephant) and *Trichechus manatus latirostris* (Florida manatee), a highly conserved region in other sequences. Because of the close taxonomic relationship between these two organisms, and the fact that their sequence was assembled on the same Broad Institute platform as many other species in our alignment that lack the deletion, we assessed this gap as likely real.

We constructed two PMSAs. Our goal was to create a curated PMSA that would optimize predictions for pathogenicity of variants from computational algorithms. This 10-sequence PMSA contained species chosen to reflect as closely as possible the 14-species PMSA previously reported for analyzing variants and validating computational algorithms in the MMR genes, in which missense VUS are common and *in silico* interpretation is frequently used [7]. We identified full length APC sequences for 11 of these 14 species. The 10-species PMSA that we curated using the above criteria (Table 1, PMSA excerpt in Fig 1, full PMSA in S1 Fig) contained five mammalian APC sequences plus chicken (*Gallus gallus*), frog (*Xenopus laevis*),

**Table 1. APC amino acid sequences from the NCBI database used in the ten species APC Protein Multiple Sequence Alignment (PMSA) and phylogenetic tree.**

| Species | APC |
|---------|-----|
| Human (*Homo sapiens*) | AAA03586.1 |
| Monkey (*Macaca mulatta*) | XP_014996065.1 |
| Cow (*Bos taurus*) | NP_001069454.2 |
| Mouse (*Mus musculus*) | NP_031488.2 |
| Opossum (*Monodelphis domestica*) | XP_007497871.1 |
| Chicken (*Gallus gallus*) | XP_004949340.1 |
| Frog (*Xenopus laevis*) | NP_001084351.1 |
| Zebrafish (*Danio rerio*) | NP_001137312.1 |
| Sea urchin (*Strongylocentrotus purpuratus*) | XP_783363.3 |
| Sea squirt (*Ciona intestinalis*) | XP_018668496.1 |

zebrafish (*Danio rerio*), sea urchin (*Strongylocentrotus purpuratus*), and sea squirt (*Ciona intestinalis*). A larger PMSA with the full set of 38 full length sequences also was constructed, with reconstitution of obvious missing exons but no detailed curation (S2 Fig).

Manual curation was often necessary to identify and label correct exon-intron boundaries and address insertions, gaps, and poorly-conserved areas where the alignment was less certain. A small amount of manual curation of gaps and insertions was required for vertebrate species. The intronic regions flanking large insertions were examined and assessed as potential splice sites. Sites with a high splice score (see Methods) were interpreted as actual splice sites and retained for creation of the phylogenetic tree. Inserted sequences flanked by a lower than average splice site were omitted from further analyses.

More extensive manual curation was required for *C. intestinalis* and *S. purpuratus*, the most distant species used, to ensure an accurate alignment and tree. Using BLAST+ on insertions in

**Fig 1. Excerpt of the curated APC alignment generated from the MSA program Clustal Omega.** Exon boundaries are labeled in red with a black background. The red highlighted region in the human sequence corresponds to a portion of an Armadillo Repeat domain.

sea squirt and sea urchin that were not present in the human sequence, we identified sequences with little homology on inspection to the vertebrate APC sequences. Exon 1 (M1 to Q46) and Exon 5, 6, and 7 (A265 to K414) of sea squirt (*C. intestinalis*) and exon 6 (A260 to F477) of sea urchin *(S. purpuratus)* did not align with the other APC sequences, returned negative BLAST results, and were removed from the final PMSA. A region of *S.purpuratus* was found with homology to a spindle fiber sequence, and a long region in its C-terminus was homologous to a herpesvirus sequence. Because the exons containing these sequences also contained regions with high homology to APC, the full exons were retained in our PMSA. A large insertion in *S. purpuratus* containing many consecutive glutamines presumably represents a coding region microsatellite. Sequences flanking this insertion were found with high splice scores, so it was kept in the alignment.

## Evolutionary rate of APC

To predict if a given invariant position is invariant with statistical significance (>95% probability), the PMSA must contain >3.0 substitutions/site [5, 6]. In addition to our ten-sequence PMSA, curated alignments were created of nine and eight sequences that omitted the more distant species *Ciona intestinalis* (sea squirt) and *Strongylocentrotus purpuratus* (sea urchin) (S3 and S4 Figs). Applying the PHYLIP ProtPars package to the curated 8, 9, and 10 species APC PMSAs, we calculated that our ten species curated *APC* alignment contained 3.3 substitutions per site (subs/site), sufficient for proceeding with subsequent analyses (see Methods). Both eight- and nine-sequence PMSAs, omitting the nonvertebrate species, contained fewer than three subs/site. We calculated subs/site for six other PMSAs of cancer susceptibility genes found on the Align-GVGD website using the same 10 species (Table 2). *APC* had a comparable evolutionary rate with *CHEK2* and *PMS2*, whereas three MMR genes (*MLH1*, *MSH2*, *MSH6*) were better conserved (1.6–2.1 subs/site), and *RAD51* was the most well-conserved of the seven genes (0.62 subs/site).

## Phylogenetic tree construction

Phylogenetic trees were generated using Bayesian, Maximum Likelihood, and Maximum Parsimony -based methods. The methods yielded similar trees, and the Maximum Parsimony -based examples are displayed in Fig 2A (10 species) and 2B (38 species). The relationships of the *APC* sequences among different species was as expected with sea urchin and sea squirt as the most distantly related organisms to humans.

**Table 2. Substitutions per site in PMSAs of seven hereditary cancer genes.**

| Protein | Substitutions per site |
|---------|------------------------|
| PMS2 | 3.4 |
| APC | 3.3 |
| CHEK2 | 3.2 |
| MSH6 | 2.8 |
| MLH1 | 2.1 |
| MSH2 | 1.6 |
| RAD51 | 0.62 |

Calculations using 10 species listed in Table 1, with evolutionary depth to sea squirt, using the PHYLIP ProtPars package.

(A)

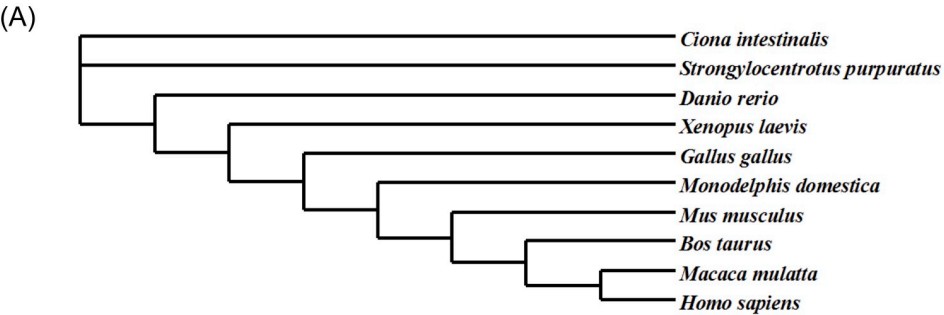

(B)

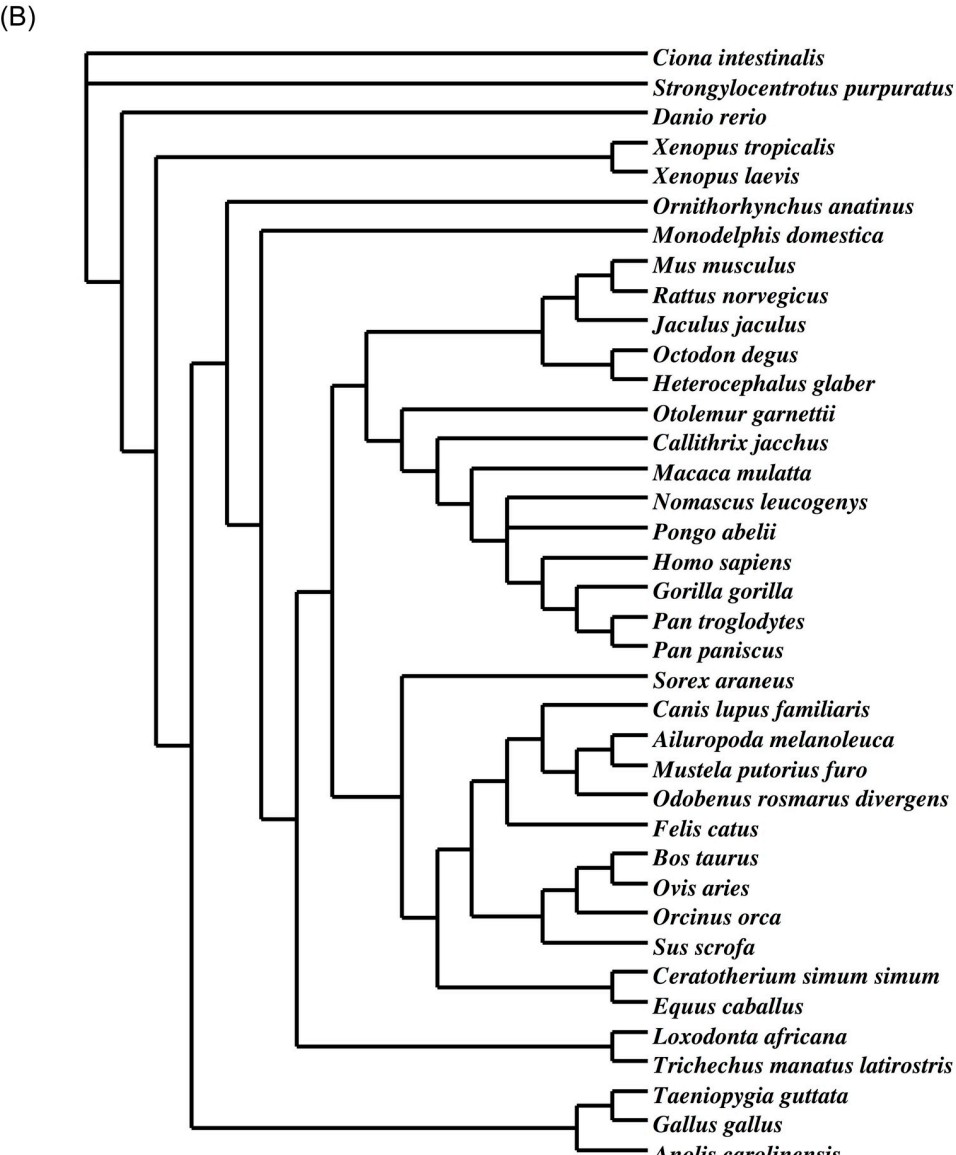

**Fig 2.** A. Ten species phylogenetic consensus tree for the APC protein constructed using the computational phylogenetics program PAUP* (Phylogenetic Analysis Using Parsimony *and other methods). B. Thirty-eight species phylogenetic consensus tree for the APC protein constructed using the computational phylogenetics program PAUP* (Phylogenetic Analysis Using Parsimony *and other methods).

**Table 3. Nine APC missense variants using filters for "missense, pathogenic, likely pathogenic".**

| APC Classified Pathogenic Variant | ClinVar Classification | Type of Variant |
|---|---|---|
| R141S | Pathogenic | Splice Site |
| K516N | Pathogenic | Splice Site |
| K581N | Likely Pathogenic | Splice Site |
| S634R | Likely Pathogenic | Exonic Splice Enhancer site |
| R653M | Pathogenic | Splice Site |
| R653G | Pathogenic | Splice Site |
| R653K | Pathogenic | Splice Site |
| G1120E | Pathogenic | Somatic |
| S1395C | Pathogenic | Somatic |

Two putative Pathogenic variants are due to somatic mutations, six are located in canonical splice sites and one occurs within an Exonic Splicing Enhancer sequence.

## *APC* variants from public databases

In the LOVD database maintained by the International Society for Gastrointestinal Hereditary Tumors (InSiGHT), in July 2013 there were a total of 46 *APC* missense variants. In ClinVar in July 2018, there were a total of 4891 *APC* variants, of which 1988 are missense. Using filters of "missense, pathogenic, likely pathogenic", yielded nine variants in the ClinVar database with assertions of Pathogenic/Likely Pathogenic (P/LP) and no conflicting interpretations of pathogenicity per ClinVar criteria. Upon further examination, it was determined that two variants were somatic mutations, and the pathogenicity of the other seven variants were inferred to be from a splicing abnormality. Six were found to occur at canonical splice sites, and the seventh occurs within an Exonic Splicing Enhancer sequence, with confirming RNA and *in vitro* evidence of splicing alterations [26] (Table 3). Thus, no pathogenic missense germline *APC* variants were documented in ClinVar using these search parameters. There are n = 21 variants (1.3% of all missense variants) with assertions of Benign or Likely Benign (B/LB). All of these were classified using criteria other than *in silico* algorithms. Of the remaining variants in Clin-Var, 93.5% of the missense variants are reported as "Unknown Significance"; the rest are classified as either "Other", or display conflicting assertions of pathogenicity (Table 4).

## Computational methods to classify APC variants

To predict the pathogenic effects of missense substitutions, multiple computational algorithms based on PMSAs and evolutionary conservation have been developed. We applied five of these tools (SIFT, PolyPhen2, Align-GVGD, MAPP, REVEL) to analyze *APC* missense variants.

**Table 4. Proportion of APC missense variants from the NCBI ClinVar database with each clinical significance classification.**

| ClinVar "Clinical Significance" for APC | Missense Variants (N = 1924) |
|---|---|
| Benign/Likely Benign | 21 (1.1%) |
| Pathogenic/Likely Pathogenic | 0 (0%) |
| Uncertain Significance | 1800 (93.5%) |
| Conflicting Interpretations of Pathogenicity | 103 (5.4%) |

Substitutions flanking the 12 splice sites found in Human APC were removed from the list of selected missense variants. A total of 1924 variants that met the above classification criteria and were not located in exon boundaries were used for analysis. Of the 1924 variants, 1.1% were classified as benign, none were classified as pathogenic and 98.9% were classified as uncertain or conflicting interpretation of pathogenicity.

**Table 5. Predictions of substitution severity with different *in silico* programs.**

| Method | Classification | Benign Variants (N = 21) | | | VUS (N = 1904) |
|---|---|---|---|---|---|
| | | Total (%) | Specificity | Total Accuracy | Predictions: Total (%) |
| ClinVar | Pathogenic | 0 (0%) | - | - | - |
| | Benign | 21 (100%) | | | - |
| REVEL | Deleterious (REVEL score ≥ 0.5) | 0 (0%) | 100% | 100% | N/A |
| | Neutral (REVEL score < 0.5) | 21 (100%) | | | N/A |
| A-GVGD | Class C65 (Deleterious moderate) | 0 (0%) | 100% | 100% | **77 (4.0%)** |
| | Class C55 (Deleterious supporting) | 0 (0%) | | | **37 (1.9%)** |
| | Class C45 (Deleterious supporting) | 0 (0%) | | | **8 (0.42%)** |
| | Class C35 (Deleterious supporting) | 0 (0%) | | | **27 (1.4%)** |
| | Class C25 (Deleterious supporting) | 0 (0%) | | | **64 (3.3%)** |
| | Class C15 (Deleterious supporting) | 0 (0%) | | | **120 (6.3%)** |
| | Class C0 (Neutral) | 21 (100%) | | | **1571 (82.5%)** |
| SIFT | Deleterious | 1 (4.8%) | 95.4% | 95.2% | **608 (31.9%)** |
| | Tolerated | 20 (95.2%) | | | **1296 (68.1%)** |
| PolyPhen2 | Probably Damaging | 1 (4.8%) | 84.0% | 80.9% | **814 (42.8%)** |
| | Possibly Damaging | 3 (13.3%) | | | **309 (16.2%)** |
| | Benign | 17 (80.9%) | | | **781 (41.0%)** |
| MAPP | Pathogenic (MAPP score ≥ 4.5) | 5 (23.8%) | 80.7% | 76.2% | **1428 (75.0%)** |
| | Neutral (MAPP score < 4.5) | 16 (76.2%) | | | **476 (25.0%)** |

Predictions of pathogenicity for APC missense variants were made using REVEL, A-GVGD, SIFT, PolyPhen2 and MAPP. REVEL output classes were designated as "Deleterious" for variants with a REVEL score ≥ 0.5 and "Neutral" with a REVEL score < 0.5 [25]. Assigning A-GVGD output Classes as "Neutral", "Deleterious moderate" and "Deleterious supporting" are based on probabilities from [27] and quantitative modeling of the ACMG/AMP criteria for assigning pathogenicity [13, 28]. SIFT predicts substitutions with SIFT scores less than 0.05 as "Deleterious" and scores equal to or greater than 0.05 as "Tolerated" [22]. PolyPhen2 predicts variants based on a Position Specific Independent Count (PSIC) score as "Benign" and "Probably Damaging" with high confidence, while a prediction of "Possibly Damaging" is predicted to be damaging, but with low confidence [23]. For MAPP, we used a cutoff score of 4.5 to predict "Pathogenic" versus "Neutral" substitutions based the cutoff used to distinguish pathogenic and neutral variants for MLH1 and MSH2 [7].

For the n = 21 variants classified in ClinVar as B/LB, the prediction algorithms showed good concordance with each other and with the ClinVar classifications (Table 5). REVEL and A-GVGD showed 100% concordance with ClinVar, SIFT predicted 95.5%, PolyPhen2 81.8%, and MAPP 77.8% to be Neutral. For the n = 1904 variants classified as VUS, "Other", or conflicting, the output differed significantly among the four non-aggregating methods (excluding REVEL). The proportion of variants predicted to be "Benign" were MAPP 25.0%, PolyPhen2 41.0%, SIFT 68.1%, Align-GVGD 82.5% (Table 4). For MAPP, we initially used the cutoff score of 4.5 previously established to distinguish P/LP from B/LB *MLH1* and *MSH2* variants [7]. This cutoff predicted 75% of APC VUS to be pathogenic, an improbable proportion. With no known pathogenic missense variants, it is unclear what cutoff score is appropriate. The lowest MAPP cutoff score (34.79) that achieved a specificity and total accuracy of 100% for classifying benign variants predicts 2.6% of VUS as pathogenic.

We explored the hypothesis that protein structural features would be associated with the likelihood that a VUS was pathogenic or benign. APC contains multiple repeats of the β-catenin binding and armadillo repeats, plus domains for oligomerization, and binding to microtubules, and EB1 and DLG proteins [29]. We hypothesized that missense variants 1) in the β-catenin binding and armadillo repeats would be neutral, since there was domain redundancy, 2) in the non-repeated domains would be more likely to be pathogenic, and 3) in unstructured regions would be neutral. There was no difference in the distribution of variants classified in

**Table 6. Proportion of benign/likely benign variants and variants of unknown significance by APC protein structural feature.**

| Domain | Benign/Likely Benign | Unknown Significance |
|---:|---:|---:|
| Beta catenin | 5 (23.8%) | 606 (31.8%) |
| Armadillo | 1 (4.8%) | 156 (8.2%) |
| Other domains | 4 (19.0%) | 378 (19.9%) |
| Not in domain | 11 (52.4%) | 764 (40.1%) |
| Total | 21 (100%) | 1904 (100%) |

ClinVar as neutral versus VUS relative to the beta catenin, armadillo, or other domains (Table 6).

Per our examination of the ClinVar database in May 2018, all APC missense mutations noted as P/LP were found to be somatic mutations, or located in canonical splice sites, or located in Exonic Splicing Enhancer sequences. Shortly after we closed our data set, p.S1028N, located in the first of four highly conserved 15-amino acid repeats within the β-catenin binding domain, was submitted to ClinVar by Ambry Genetics and classified as Likely Pathogenic. The evidence for this classification includes, as per the ACMG/AMP guidelines, segregation score (PP1_Strong, six meioses), phenotype score (PS4_Moderate), functional domain (PM1 [30]), population frequency score (PM2_Supporting) and *in silico* data (PP3). There is no evidence of splice abnormality. This variant would reach LP regardless of *in silico* analysis. Further scrutiny of variants in this region demonstrates one other variant, p.N1026S, classified as "Conflicting Interpretations of Pathogenicity" in ClinVar, which satisfies the ACMG/AMP guidelines as LP. The same criteria (PP1_Strong, PS4_Moderate, PM1, PM2) can be applied to p.N1026S, in addition to a functional defect (PS3) as reported in the literature [30, 31]. N1026 and S1028 are both located in the first 15-amino acid repeat of the β-catenin binding domain and after careful review are the only LP/P *APC* missense variants that we found in ClinVar in July 2018 that satisfy the ACMG/AMP guidelines.

## Discussion

*In silico* tools have been validated with accepted standards for relatively few genes, and the field would greatly benefit from refinement of standards for applying these tools. Factors that have been shown to be important for interpreting the output and reliability of computational algorithms include quality of PMSA (reviewed in [4]), and choice of variant data sets [32]. An important factor regarding data sets that has emerged recently is how predictors should not be evaluated on variants or proteins that were used to train their prediction models. This circularity could result in predictive values that are artificially inflated [32, 33], and could occur with either likely pathogenic or likely benign variants. We suggest that not enough attention has been assigned to an additional important factor, the likelihood that missense substitution is a major mechanism of pathogenicity for a gene.

Our analysis suggests possible revisions to the ACMG/AMP classification scheme for pathogenicity, which defines multiple criteria for evidence of benign or pathogenic effect, with strength ranging from "Supporting" to "Very Strong", and rules for combining different types of evidence [13]. For example, criterion BP1, "Missense variant in a gene for which primarily truncating variants are known to cause disease", is relevant to *APC*. By this criterion, any missense APC variant is given "Supporting" evidence, the lowest level, favoring benign classification of missense variants. Further study may help determine whether this criterion for benign classification should be upgraded from "Supporting" (for which estimated Odds of Pathogenicity is low [28], discussed below) to a higher level for these variants. The PP2 criterion for

pathogenicity presupposes that missense is a common mechanism for mutation; future studies should assess whether it is being inappropriately used when missense is a rare or unknown mechanism for a given gene.

Our work confirms that PMSA construction remains a labor-intensive task [34]. Current automated tools do not align unstructured regions accurately, resulting in errors that require manual curation of protein and nucleotide sequences in order to optimally curate a full alignment. For many genes, accurate PMSA can prove important for *in silico* analysis of variant pathogenicity [4]. There is no consensus in the assessment of PMSA quality, although metrics have been proposed [35]. We and others have proposed that a PMSA should include enough sequences to contain three subs/site in order for predictions to be statistically robust [5, 6], and for APC we achieved this threshold with the addition of non-vertebrate sequences. We chose our sequences to be consistent with PMSAs of other cancer susceptibility genes for which *in silico* algorithms have proven to be valuable tools for variant classification. PMSAs for 15 such genes are posted on the Align-GVGD (http://agvgd.hci.utah.edu/about.php) web site. We hope to promote standardization of methods for the purposes of *in silico* analysis for variant classification. It remains to be determined whether a consistent set of sequences will be most appropriate for other gene sets. The creation and validation of our APC PMSA did identify interesting features of gene evolution and of genome annotation and analysis, and we anticipate that PMSAs across gene families are likely to elucidate specific structure-function relationships and molecular pathways of critical cellular functions. The full APC PMSA can be seen in S1 Fig, where it can be used for purposes that are beyond the scope of this paper.

One cannot assume that *in silico* tools that are valuable predictors for one gene will perform as well for other genes. The majority of *APC* missense variants in ClinVar are likely to be benign, given the paucity of missense pathogenic variants identified in over two decades of clinical *APC* testing. An example of a similar gene is *CDH1*, in which pathogenic missense variants also are rare. An expert panel studying the *CDH1* gene has recommended that computational methods not be used for missense *CDH1* variants [36]. Thus, tools that work well for genes that are commonly inactivated by missense changes [7, 11, 37, 38] can be misleading for genes that are rarely inactivated by missense. For such genes, traditional *in silico* tools will likely overestimate the probability of pathogenicity of any missense variant.

The ClinGen Sequence Variant Interpretation working group has estimated that the "Supporting" level of evidence confers approximately 2.08/1 odds in favor of pathogenicity [28], or a 67.5% probability of pathogenicity. Our current analyses of APC variants suggest that the likelihood that a missense APC variant is pathogenic is very low, perhaps lower than 1%. We base this conclusion on the observation that only 2 of 1924 missense variants in Clinvar (0.10%) are classified as pathogenic (Table 3) after decades of analysis by testing labs and researchers. Despite this, our curated APC PMSA and several *in silico* prediction tools all predicted a significant fraction of missense variants to be pathogenic (Table 4). The methods that we used varied widely in their predictions for APC VUS; predictions of Pathogenic or Likely Pathogenic ranged from 17.5% to 75%, all of which are higher than the likely figure by at least an order of magnitude. This provides mathematical support for not using *in silico* evidence in favor of pathogenicity (PP3 in the ACMG/AMP scheme [13]) for these genes. One approach might be to create a decision tree in which a gene must meet specific criteria before *in silico* evidence is applied. More work is needed in order to understand which genes require precuration to assess whether PMSA-based or other *in silico* methods are likely to be useful. A difference between functional or structural relevance to the protein and clinical relevance may occur if the assayed function is not crucial to the phenotype, or perhaps from domain redundancy or other protein structural features.

Another important factor regarding data sets is whether the subject was being tested because of clinical suspicion, or whether broad panel testing, whole exome or whole genome sequencing yielded a variant in the absence of any known clinical features. The degree of clinical suspicion is difficult to discern from the majority of ClinVar *APC* variants. The prior probability of pathogenicity [7] will be much lower for a variant discovered incidentally through whole exome sequencing compared with one identified through clinical testing because of a strong history of polyposis and/or colon cancer, with intermediate scenarios also possible.

Computational methods can be an excellent classifier for missense variants in hereditary cancer genes where missense is a common mechanism of pathogenicity [7–11]. However, known pathogenic APC missense germline variants are rare. It is possible that none exist outside of the first 15-amino acid repeat of the β-catenin binding domain, and it is unknown how many other pathogenic missense variants are located in this 15 amino acid repeat, complicating the use of computational tools. Further analysis of this region is necessary to understand the role of missense APC variants and the value of *in silico* algorithms. The β-catenin binding repeats may be the only specific region of 15 AA out of the 2843 AA of APC in which *in silico* methods may be predictive of clinical pathogenicity. A similar observation to the use of *in silico* analysis has been made regarding the BRCT domain of BRCA1 [4]. There may be characteristics of the APC gene and protein that confound the results of *in silico* algorithms. One plausible hypothesis for the failure of missense variants to abrogate APC function is the redundancy of APC important structural elements (armadillo repeats, β-catenin and axin binding sites) [29], so the inactivation of a single repeat might not eliminate binding to the target to a clinically relevant level.

Defining features that distinguish genes for which missense is a common (e.g., MMR genes [7]) versus uncommon (e.g., *CDH1* [36, 39], *RB1* [40]) pathogenic mechanism would significantly improve the application of *in silico* tools to variant classification. We propose that *in silico* methods to assess missense variants (PP3 and BP4 in the ACMG/AMP guidelines [13]) be used sparingly for any gene where strong evidence suggests that missense rarely causes pathogenicity. Future work might consider whether BP4 (concordance for "benign" classification among multiple methods) might be replaced by BP1 (truncation predominates, missense unlikely) in such cases. Our results suggest that a systematic study of variant pathogenicity and protein features such as domain structure is warranted to improve the use of predictive algorithms in hereditary cancer genes.

## Supporting information

**S1 Fig. Curated 10-species APC alignment.** PMSA was generated from the program Clustal Omega. Exon boundaries are labeled in red with a black background. The domains are highlighted throughout the alignment. Grey is oligomerization domain, red is Armadillo repeats, yellow is Beta Catenin Repeats, green is a sequence with homology to the herpes virus (PHA03307), turquoise is the Basic domain, and purple is the EB1 and HDLG binding site.
(PDF)

**S2 Fig. 38-species APC alignment.** PMSA was generated from the program Clustal Omega. No annotation is added.
(PDF)

**S3 Fig. Curated 9-species APC alignment.** PMSA was generated from the program Clustal Omega. Annotation as per S1 Fig.
(PDF)

**S4 Fig. Curated 8-species APC alignment.** PMSA was generated from the program Clustal Omega. Annotation as per S1 Fig.
(PDF)

# Acknowledgments

We are grateful for ongoing illuminating discussions with our colleagues on the ClinGen Sequence Variant Interpretation Working group and the InSiGHT Variant Interpretation Committee.

# Author Contributions

**Conceptualization:** Indra Neil Sarkar, Marc S. Greenblatt.

**Data curation:** Alexander D. Karabachev, Dylan J. Martini, David J. Hermel, Dana Solcz, Marcy E. Richardson, Tina Pesaran.

**Formal analysis:** Dylan J. Martini, David J. Hermel, Dana Solcz, Marcy E. Richardson, Tina Pesaran, Indra Neil Sarkar, Marc S. Greenblatt.

**Investigation:** Alexander D. Karabachev, Dylan J. Martini, David J. Hermel, Dana Solcz, Indra Neil Sarkar, Marc S. Greenblatt.

**Methodology:** Indra Neil Sarkar, Marc S. Greenblatt.

**Project administration:** Marc S. Greenblatt.

**Software:** Indra Neil Sarkar.

**Writing – original draft:** Alexander D. Karabachev, Dylan J. Martini, David J. Hermel, Dana Solcz, Indra Neil Sarkar, Marc S. Greenblatt.

**Writing – review & editing:** Alexander D. Karabachev, David J. Hermel, Dana Solcz, Marcy E. Richardson, Tina Pesaran, Indra Neil Sarkar, Marc S. Greenblatt.

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
