## [Decision Letter · Decision Letter 0]

2 Mar 2020

PONE-D-19-28042

Curated multiple sequence alignment for the Adenomatous Polyposis Coli (APC) gene and accuracy of In Silico pathogenicity predictions

PLOS ONE

Dear Dr. Greenblatt,

Thank you for submitting your manuscript to PLOS ONE. After careful consideration, we feel that it has merit but does not fully meet PLOS ONE’s publication criteria as it currently stands. Therefore, we invite you to submit a revised version of the manuscript that addresses the points raised during the review process.

Theauthors have showed that with PMSA they have reassigned 5/21 to be pathogenic, they should mention those 5 variants and discuss how this differs from the others and the location and consequences of the variants in addition to  addressing the points raised by reviewer 1.

We would appreciate receiving your revised manuscript by Apr 11 2020 11:59PM. To enhance the reproducibility of your results, we recommend that if applicable you deposit your laboratory protocols in protocols.io, where a protocol can be assigned its own identifier (DOI) such that it can be cited independently in the future. For instructions see: http://journals.plos.org/plosone/s/submission-guidelines#loc-laboratory-protocols

We look forward to receiving your revised manuscript.

Kind regards,

Obul Reddy Bandapalli, MSc, PhD

Academic Editor

PLOS ONE

Journal Requirements:

"I have read the journal's policy and the authors of this manuscript have the following competing interests: MER and TP are employees of Ambry Genetics, Inc. No other authors have competing interests."

We note that one or more of the authors are employed by a commercial company: Ambry Genetics.

Reviewers' comments:

Reviewer's Responses to Questions

**Comments to the Author**

1. Is the manuscript technically sound, and do the data support the conclusions?

Reviewer #1: Yes

Reviewer #2: Yes

2. Has the statistical analysis been performed appropriately and rigorously? 

Reviewer #1: N/A

Reviewer #2: Yes

3. Have the authors made all data underlying the findings in their manuscript fully available?

Reviewer #1: Yes

Reviewer #2: Yes

4. Is the manuscript presented in an intelligible fashion and written in standard English?

Reviewer #1: No

Reviewer #2: Yes

5. Review Comments to the Author

Reviewer #1: Prediction of pathogenicity of variants of unknown significance is a clinically important goal and in this study the authors have constructed a PMSA of APC to assist in variant prediction. Not all the steps in the description of how they made the PMSA are completely clear but overall it looks like the authors located as many complete APC protein sequences they could find, did a multiple sequence alignment and adjusted the MSA according to their best judgement as to how the sequence should align. Having constructed the PMSA 5 algorithms were applied that predict impact of AA substitutions to a PMSA they constructed for APC to see what they predicted at sites in APC listed as pathogenic or benign in ClinVar and scored how well each algorithm matched the ClinVar annotations.

As a non-bioinformatics specialist I found the paper very hard to follow and was not able to assess to my satisfaction the reliability of the variant prediction using the newly constructed PMSA. For example online 403 in the discussion the statement is made "Our current analyses of APC variants suggest that the likelihood that a missense APC variant is pathogenic is far lower than 1%." However I could not find any reference to this in the results so have no idea how this figure was derived. While this conclusion may well be in the results somewhere, it is not obvious and I suspect many readers of PLOS One will have difficulty following the flow of the manuscript. I suggest the authors should revise the manuscript to more explicitly explain the various steps in the analyses and how they reach their conclusions.

Reviewer #2: This is one of only two papers I have ever reviewed that I have recommended acceptance as is. It is excellent and a major advance in the field. The recommendation in the discussion that the ACMG/AMP criteria re APC variants needs amending is especially germane, and will be relevant to genes with similar modes of pathogenicity.

6. PLOS authors have the option to publish the peer review history of their article (what does this mean?). If published, this will include your full peer review and any attached files.

Reviewer #1: No

Reviewer #2: Yes: Ian Frayling

---

## [Author Response · Author response to Decision Letter 0]

1 Apr 2020

We thank the reviewers and editor for their helpful comments. We are very happy that Reviewer 2 took such a positive view of the paper. Reviewer 1 and the editor each make one specific suggestion, which we have addressed as below. Also, we now provide the “data not shown”.

1. Reviewer 1 couldn’t follow the conclusion that "Our current analyses of APC variants suggest that the likelihood that a missense APC variant is pathogenic is far lower than 1%." We have modified this statement slightly, and provided the rationale, which is already in the manuscript on pages 11 and 12, and in Table 3. We think we have made this more clear:

"Our current analyses of APC variants suggest that the likelihood that a missense APC variant is pathogenic is very low, perhaps lower than 1%. We base this conclusion on the observation that only 2 of 1924 missense variants in Clinvar (0.10%) are classified as pathogenic (Table 3) after decades of analysis by testing labs and researchers." 

2. The specific comment from the editor is slightly puzzling, but we think that we have found the source of the comment and edited the manuscript to clarify: “The authors have showed that with PMSA they have reassigned 5/21 to be pathogenic, they should mention those 5 variants and discuss how this differs from the others and the location and consequences of the variants in addition to addressing the points raised by reviewer 1.”

We believe that the reviewer is referring to the sentence “When applied to 21 variants reported in ClinVar as Benign, the five methods ranged in accuracy from 76.2-100%” in the Abstract. We have clarified that we consider the variants to be securely classified as Benign. These variants are described in the manuscript on pages 12, 13, and 14, and in Tables 4, 5A, and 5B. We have edited sentences in the Abstract and in the text (p.13, line 292-294) to clarify that we consider these to be securely classified as Benign, and the discrepancy is due to the methods, not due to misclassification, as follows:

Abstract: “When applied to 21 missense variants reported in ClinVar and securely classified as Benign, the five methods ranged in accuracy from 76.2-100%.”

Manuscript text: “For the n=21 variants securely classified in ClinVar as B/LB based on non-computational data, the prediction algorithms showed good concordance with each other and with the ClinVar classifications (Table 5A). “ 

3. Data Not Shown 

The “Data not shown” were the 8 sequence and 9 sequence alignments. These are now included as Supporting Figure 3 and 4.

4. We have edited the Competing Interests section using the language provided in the Review.

---

## [Editor Report · Decision Letter 1]

12 May 2020

Curated multiple sequence alignment for the Adenomatous Polyposis Coli (APC) gene and accuracy of In Silico pathogenicity predictions

PONE-D-19-28042R1

Dear Dr. Greenblatt,

We are pleased to inform you that your manuscript has been judged scientifically suitable for publication after internal evaluation of  the revised manuscript and found that the authors addressed the points raised by the reviewer/editor and found them to be satisfactory and will be formally accepted for publication once it complies with all outstanding technical requirements.

With kind regards,

Obul Reddy Bandapalli, MSc, PhD

Academic Editor

PLOS ONE
---

## [Editor Report · Acceptance letter]

22 Jul 2020

PONE-D-19-28042R1 

Curated multiple sequence alignment for the Adenomatous Polyposis Coli (*APC*) gene and accuracy of *In Silico* pathogenicity predictions 

Dear Dr. Greenblatt:

I'm pleased to inform you that your manuscript has been deemed suitable for publication in PLOS ONE. Congratulations! Your manuscript is now with our production department. 

Kind regards, 

on behalf of

Dr. Obul Reddy Bandapalli 

Academic Editor

PLOS ONE